# Generating Wikipedia by Summarizing Long Sequences

**Peter J. Liu**[*]**, Mohammad Saleh**,[*]
**Etienne Pot**[†]**, Ben Goodrich, Ryan Sepassi, Łukasz Kaiser, Noam Shazeer**
Google Brain
Mountain View, CA
{peterjliu,msaleh,epot,bgoodrich,rsepassi,lukaszkaiser,noam}@google.com

## Abstract

We show that generating English Wikipedia articles can be approached as a multi-document summarization of source documents. We use extractive summarization to coarsely identify salient information and a neural abstractive model to generate the article. For the abstractive model, we introduce a decoder-only architecture that can scalably attend to very long sequences, much longer than typical encoder-decoder architectures used in sequence transduction. We show that this model can generate fluent, coherent multi-sentence paragraphs and even whole Wikipedia articles. When given reference documents, we show it can extract relevant factual information as reflected in perplexity, ROUGE scores and human evaluations.

## 1 Introduction

The sequence-to-sequence framework has demonstrated success in natural-language sequence transduction tasks such as machine translation. More recently, neural techniques have been applied to do single-document, abstractive (paraphrasing) text summarization of news articles (Rush et al. (2015), Nallapati et al. (2016)). In this prior work, the input to supervised models ranged from the first sentence to the entire text of an article, and they are trained end-to-end to predict reference summaries. Doing this end-to-end requires a significant number of parallel article-summary pairs since language understanding is a pre-requisite to generate fluent summaries.

In contrast, we consider the task of multi-document summarization, where the input is a collection of related documents from which a summary is distilled. Prior work has focused on extractive summarization, which select sentences or phrases from the input to form the summaries, rather than generating new text. There has been limited application of abstractive neural methods and one possible reason is the paucity of large, labeled datasets.

In this work, we consider English Wikipedia as a supervised machine learning task for multi-document summarization where the input is comprised of a Wikipedia topic (title of article) and a collection of non-Wikipedia reference documents, and the target is the Wikipedia article text. We describe the first attempt to abstractively generate the first section, or *lead*, of Wikipedia articles conditioned on reference text. In addition to running strong baseline models on the task, we modify the Transformer architecture (Vaswani et al., 2017) to only consist of a decoder, which performs better in the case of longer input sequences compared to recurrent neural network (RNN) and Transformer encoder-decoder models. Finally we show our modeling improvements allow us to generate entire Wikipedia articles.

---

[*]Joint first-authors. Ordered randomly.
[†]Work done as a member of the Google Brain Residency (g.co/brainresidency)

## 2 RELATED WORK

### 2.1 OTHER DATASETS USED IN NEURAL ABSTRACTIVE SUMMARIZATION

Neural abstractive summarization was pioneered in Rush et al. (2015), where they train headline generation models using the English Gigaword corpus (Graff & Cieri, 2003), consisting of news articles from number of publishers. However, the task is more akin to sentence paraphrasing than summarization as only the first sentence of an article is used to predict the headline, another sentence. RNN-based encoder-decoder models with attention (seq2seq) perform very well on this task in both ROUGE (Lin, 2004), an automatic metric often used in summarization, and human evaluation (Chopra et al., 2016).

In Nallapati et al. (2016), an abstractive summarization dataset is proposed by modifying a question-answering dataset of news articles paired with story highlights from Daily Mail and CNN. This task is more difficult than headline-generation because the information used in the highlights may come from many parts of the article and not only the first sentence. One downside of the dataset is that it has an order-of-magnitude fewer parallel examples (310k vs. 3.8M) to learn from. Standard seq2seq models with attention do less well, and a number of techniques are used to augment performance. Another downside is that it is unclear what the guidelines are for creating story highlights and it is obvious that there are significant stylistic differences between the two news publishers.

In our work we also train neural abstractive models, but in the multi-document regime with Wikipedia. As can be seen in Table 1, the input and output text are generally much larger, with significant variance depending on the article. The summaries (Wikipedia lead) are multiple sentences and sometimes multiple paragraphs, written in a fairly uniform style as encouraged by the Wikipedia Manual of Style[1]. However, the input documents may consist of documents of arbitrary style originating from arbitrary sources.

We also show in Table 1 the ROUGE-1 recall scores of the output given the input, which is the proportion of unigrams/words in the output co-occuring in the input. A higher score corresponds to a dataset more amenable to extractive summarization. In particular, if the output is completely embedded somewhere in the input (e.g. a wiki-clone), the score would be 100. Given a score of only 59.2 compared to 76.1 and 78.7 for other summarization datasets shows that ours is the least amenable to purely extractive methods.

### 2.2 TASKS INVOLVING WIKIPEDIA

There is a rich body of work incorporating Wikipedia for machine learning tasks, including question-answering (Hewlett et al. (2016), Rajpurkar et al. (2016)) and information extraction (Lehmann et al., 2015), and text generation from structured data (Lebret et al., 2016).

The closest work to ours involving generating Wikipedia is Sauper & Barzilay (2009), where articles are generated extractively (instead of abstractively in our case) from reference documents using learned templates. The Wikipedia articles are restricted to two categories, whereas we use all article types. The reference documents are obtained from a search engine, with the Wikipedia topic used as query similar to our search engine references. However we also show results with documents only found in the References section of the Wikipedia articles.

### 2.3 TRANSFORMER MODELS

Previous work on neural abstractive summarization relies on RNNs as fundamental modules, mirroring techniques successful in machine translation (MT). Recently, state-of-the-art MT results were obtained using a non-recurrent architecture, called the Transformer (Vaswani et al., 2017). The lack of recurrence enables greater within-training-example parallelization, at the cost of quadratic complexity in the input sequence length. We find the Transformer transfers well to medium length, input sequence summarization and describe modifications to better handle longer sequences.

---

[1] `https://en.wikipedia.org/wiki/Wikipedia:Manual_of_Style`

Table 1: Order of magnitude input/output sizes and unigram recall for summarization datasets.

| Dataset | Input | Output | # examples | ROUGE-1 R |
|---|---|---|---|---|
| Gigaword (Graff & Cieri, 2003) | $10^1$ | $10^1$ | $10^6$ | 78.7 |
| CNN/DailyMail (Nallapati et al., 2016) | $10^2$–$10^3$ | $10^1$ | $10^5$ | 76.1 |
| WikiSum (ours) | $10^2$–$10^6$ | $10^1$–$10^3$ | $10^6$ | 59.2 |

Table 2: Percentiles for different aspects of WikiSum dataset. Size is in number of words.

| Percentile | 20 | 40 | 50 | 60 | 80 | 100 |
|---|---|---|---|---|---|---|
| Lead Size | 37 | 62 | 78 | 98 | 166 | 10,034 |
| Num Citations | 1 | 2 | 2 | 3 | 5 | 1,029 |
| Citations Size | 562 | 1,467 | 2,296 | 3,592 | 10,320 | 6,159,463 |
| Num Search Results | 10 | 20 | 26 | 31 | 46 | 2,095 |
| Search Results Size | 1,1691 | 33,989 | 49,222 | 68,681 | 135,533 | 5,355,671 |

## 3 ENGLISH WIKIPEDIA AS A MULTI-DOCUMENT SUMMARIZATION DATASET

Wikipedia, being an encyclopedia, can be viewed as a collection of summaries on various topics given by their title, e.g. "Canada" or "Machine Learning". The source material to be summarized can be viewed as all reputable documents on the Web or books; however, to make the problem more tractable we consider the following subsets of all documents, $D$:

1. Cited sources: A Wikipedia article that conforms to the style guidelines should be well-supported by citations found in the *References* section of Wikipedia articles. For each article, $a_i$, we extract all text without markup from crawlable citation documents, $C_i \subset D$, to use as input to our method.

2. Web Search results: To expand the collection of reference documents, we crawl the search results from the Google search engine, using the article section titles as queries. For each query, we collect 10 result pages. From this collection we remove the Wikipedia article itself, which is often among the top results. We also remove "clones", which are detected when there is a high-level of unigram overlap with the article (details provided in A.2.1). We denote these refined search results for an article, $a_i$, as $S_i \subset D$. Similar to $C_i$, we extract only the text to use as input.

Table 2 describes overall properties of our WikiSum dataset. Many articles have few citations, motivating our supplementation of the source documents with web search results. On the other hand, citations when available, tend to be of higher-quality. When counting the total words in the entire dataset, it is orders-of-magnitude larger than previous summarization datasets.

To have consistent train/development/test data across corpus-comparison experiments, we restrict the articles to those with at least one crawlable citation. We divide the articles roughly into 80/10/10 for train/development/test subsets, resulting in 1865750, 233252, and 232998 examples respectively.

## 4 METHODS AND MODELS

Because the amount of text in input reference documents ($C_i, S_i$) can be very large (see Table 2) it is infeasible to train an end-to-end abstractive model given the memory constraints of current hardware. Hence, we first coarsely select a subset of the input using extractive summarization. The second stage involves training an abstractive model that generates the Wikipedia text while conditioning on this extraction. This two-stage process is inspired by by how humans might summarize multiple long documents: First highlight pertinent information, then conditionally generate the summary based on the highlights.

## 4.1 Extractive stage

We investigate three extractive methods from the summarization literature, along with a trivial and cheating method, to assess the importance of this stage. For each article, $a_i$ we create a ranked list of paragraphs, $\{p^i_{R_i(j)}\}$, occurring in $(C_i, S_i)$ where $R_i(j)$ is the rank of the $j$th paragraph $p^i_j$ of $(C_i, S_i)$. From this we select the first $L$ tokens as input to the second abstractive stage.

1. *Identity*: As a trivial baseline extractor, we simply use the first $L$ tokens of the input.

2. *tf-idf*: A non-trivial ranking is to consider ranking paragraphs as documents in a query-retrieval problem, where the query is the title of the article, $T(a_i)$. We compute tf-idf (Ramos et al., 2003) for the query, with respect to the documents, $\{p^i_j\}$. That is, we summate for each word in the query

$$N_w \cdot log(\frac{N_d}{N_{dw}})$$

where $N_w$, $N_d$, and $N_{dw}$ are the count of the word in the document, total number of documents, and total number of documents containing the word, respectively.

3. *TextRank* (Mihalcea & Tarau, 2004): A weighted graph is defined where text units are nodes and edges are defined by a similarity measure based on word overlap. An algorithm similar to PageRank (Page et al., 1999) is then used to compute the ranking of text units. We used paragraphs for the text units.

4. *SumBasic* (Nenkova & Vanderwende, 2005): Word frequencies in the input text are used to assign scores to words, which are in turn used to score sentences. After selecting the best scoring sentence, words in it have their scores reduced, and the process is repeated until the desired summary length is reached.

5. *Cheating* To further demonstrate the quality of extraction on the final performance, we implement a cheating extractor that ranks $\{p^i_j\}$ using recall of bigrams in the ground truth text:

$$d(p^i_j, a_i) = \frac{bigrams(p^i_j) \cap bigrams(a_i)}{bigrams(a_i)} \quad (1)$$

## 4.2 Abstractive stage

### 4.2.1 Data representation

Given the ordered paragraphs $\{p^i_{R_i(j)}\}$, we derive the raw text input simply as the concatenation of the paragraphs in order, the most relevant at the beginning, and prefixed with the title.

We then encode the text using sub-word tokenization similar to Wu et al. (2016) with a vocabulary size of 32,000 yielding tokenized input, $x_i$:

$$text_i = T(a_i) \| \{p^i_{R_i(j)}\}$$

$$tokenize(text_i) = x_i = (x^1_i, x^2_i, ..., x^{n_i}_i)$$

For various values of $L$ in experiments, we truncate the tokens to form the input sequence:

$$m^L_i = (x^1_i, ...x^{min(L,n_i)}_i)$$

For the output, we use the same vocabulary and tokenization for the Wikipedia lead text but do not do any truncation across experiments.

Next we describe the abstractive models, $W$, that learn to write articles, $a_i = W(m^L_i)$, which we treat as a sequence transduction problem from very long input sequences (up to $L = 11000$) to medium output sequences (typically less than 500).

### 4.2.2 BASELINE MODELS

As a baseline we apply the standard LSTM encoder-decoder with attention (seq2seq-att) as in Bahdanau et al. (2014) to this task. As is typical we train to optimize the maximum-likelihood objective:

$$y_i = tokenize(a_i)$$

$$\prod_{i=1}^{N} p(y_i|m_i^L)$$

A stronger, more recent baseline that we use is the non-recurrent Transformer model described in 2.3, which also has symmetric encoder and decoder modules (T-ED).

### 4.2.3 TRANSFORMER DECODER (T-D)

We introduce a simple but effective modification to T-ED for long sequences that drops the encoder module (almost reducing model parameters by half for a given hyper-parameter set), combines the input and output sequences into a single "sentence" and is trained as a standard language model.

That is, we convert a sequence-transduction example $(m^1, ..., m^n) \mapsto (y^1, ..., y^\eta)$ into the sentence $(w^1, ..., w^{n+\eta+1}) = (m^1, ..., m^n, \delta, y^1, ..., y^\eta)$, where $\delta$ is a special separator token and train a model to predict the next word given the previous ones:

$$p(w^1, ..., w^{n+\eta}) = \prod_{j=1}^{n+\eta} p(w^i|w^1, ..., w^{j-1})$$

Since the model is forced to predict the next token in the input, $m$, as well as $y$, error signals are propagated from both input and output time-steps during training. We also suspect that for monolingual text-to-text tasks redundant information is re-learned about language in the encoder and decoder. We believe this allows for easier optimization and empirically observe this with longer sequences (see Section 5.3). Note that because of the self-attention of the Transformer, when generating the next token, attention from both $m$ and $y$ are considered. At inference we provide the input sequence, $m_i$, initially, and auto-regressively generate the output, $y_i$, as normal.

### 4.2.4 TRANSFORMER DECODER WITH MEMORY-COMPRESSED ATTENTION (T-DMCA)

To re-use the terminology used to describe the Transformer, the attention is a function of a query ($Q$) and set of key ($K$) and value ($V$) pairs. To handle longer sequences, we modify the multi-head self-attention of the Transformer to reduce memory usage by limiting the dot products between $Q$ and $K$ in:

$$Attention(Q, K, V) = softmax(\frac{QK^T}{\sqrt{d_k}})V$$

**Local attention**: Sequence tokens are divided into blocks of similar length and attention is performed in each block independently. As the attention memory cost per block becomes constant, this modification allow us to keep the number of activations linear with respect to the sequence length. In our experiments, we choose to have blocks of 256 tokens.

**Memory-compressed attention**: After projecting the tokens into the query, key, and value embeddings, we reduce the number of keys and values by using a strided convolution. The number of queries remains unchanged. This modification allows us to divide the number of activations by a compression factor. In our experiments we use convolution kernels of size 3 with stride 3. In contrast to local attention layers, which only capture the local information within a block, the memory-compressed attention layers are able to exchange information globally on the entire sequence.

These modifications (see Figure 1) allow us in practice to process sequences 3x in length over the T-D model. For both local and memory-compressed attention, masking is added to prevent the queries from attending to future keys and values. Our final architecture is a 5-layer network (LMLML) alternating between local-attention (L) layers and memory-compressed attention (M) layers (in Vaswani et al. (2017) it is 6 identical layers). We also added in some experiments one mixture of experts (MoE) layer (Shazeer et al., 2017) to increase the network's capacity.

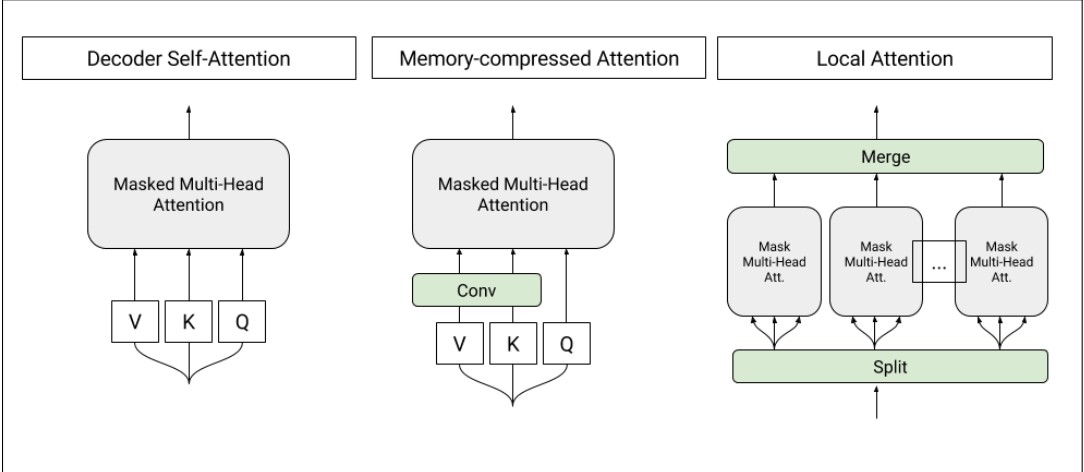

Figure 1: The architecture of the self-attention layers used in the T-DMCA model. Every attention layer takes a sequence of tokens as input and produces a sequence of similar length as the output. **Left:** Original self-attention as used in the transformer-decoder. **Middle:** Memory-compressed attention which reduce the number of keys/values. **Right:** Local attention which splits the sequence into individual smaller sub-sequences. The sub-sequences are then merged together to get the final output sequence.

## 5 EXPERIMENTS

### 5.1 EVALUATION

In experiments we evaluate based on perplexity (per-wordpiece), a common language modeling metric, and ROUGE-L F1 (version `ROUGE-1.5.5`), a common metric used in comparing candidate and reference summaries. Note the F1 flavor of ROUGE is more appropriate in this setting as we do not explicitly constrain the output length in abstractive models; it is the harmonic mean of ROUGE-Recall (which favors long summaries) and ROUGE-Precision (which favors short summaries).

Although optimizing ROUGE directly has been shown to not always yield the best summaries as evaluated by human judgment (Paulus et al., 2017), we found that for our task optimizing for perplexity correlates with increased ROUGE and human judgment. We suspect that the relatively uniform style of Wikipedia articles makes ROUGE more appropriate here than in general abstractive summarization tasks.

### 5.2 MODEL TRAINING DETAILS AND DECODING

For all abstractive model training, we use the open-source `tensor2tensor`[2] library.

The seq2seq baseline had a hidden size of 128 with 2 layers (we use the hyper-parameter set defined in the library as `lstm_attention`).

For the Transformer encoder-decoder (T-ED), we use the hyper-parameter set `transfomer_base_v1` and train for 1 million steps. Models exhibited very little over-fitting and did not require early-stopping. The Transformer Decoder (T-D) was identical to the decoder part of T-ED. The T-DMCA model is similar to T-D, but with the enhancements described in section 4.2.4.

Unless otherwise stated, during decoding we use a beam search of size 4 and length penalty $\alpha = 0.6$ (Wu et al., 2016) and decode until an end-of-sequence token is reached.

---

[2]`https://github.com/tensorflow/tensor2tensor`

Table 3: Comparison of extractive method and corpus with $L = 500$, and the Transformer E-D model

| Extractor | Corpus | Test log-perplexity | ROUGE-L |
|-----------|--------|--------------------|---------|
| *cheating* | combined | 1.72975 | 59.3 |
| *tf-idf* | combined | 2.46645 | 34.2 |
| *tf-idf* | citations-only | 3.04299 | 22.6 |
| *tf-idf* | search-only | 3.56593 | 2.8 |
| *identity* | combined | 4.80215 | 4.0 |

## 5.3 RESULTS AND DISCUSSION

There are four main dimensions we vary in experiments in generating Wikipedia lead sections:

1. Extractive method: *SumBasic*, *TextRank*, *tf-idf*, *identity*, *cheating extractor*

2. Input corpus: *citations*, *search results*, *combined*

3. Abstractive model input length, $L$: We try values between 100 and 11000.

4. Abstractive model architecture: *seq2seq-att*, *T-ED*, *T-D*, *T-DMCA*

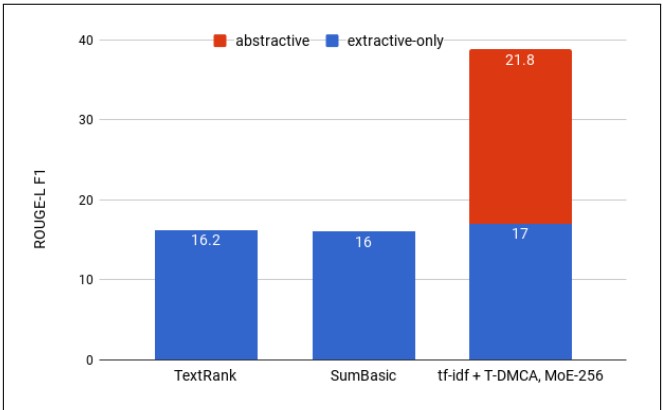

Figure 2: ROUGE-L F1 for various extractive methods. The abstractive model contribution is shown for the best combined *tf-idf*-T-DMCA model.

**Extractive-only is not enough:** We investigate performance of extractive methods without the abstractive model by looking at the ROUGE-L F1 scores after running *tf-idf*, *SumBasic*, and *TextRank* in Figure 2, without any abstractive model. In the case of TextRank and SumBasic we matched the output length to the target length and observe the extractive methods perform roughly in-line with each other in terms of ROUGE-L F1. Our best abstractive model more than doubled this metric. Further, this model yields large improvements in perceived linguistic quality (elaborated below).

**Extractive method:** From Table 3 we observe that smart extraction is critical for final abstractive performance. There is a significant gap between doing nothing, *identity*, and extractive summarization, *tf-idf*. Further, there is a significant gap between *tf-idf* and the *cheating* extractor, suggesting future work in improving the extraction step could result in significant improvements. One possibility is to train a supervised model to predict relevance (Eq. 1), which we leave as future work. For subsequent experiments we fix the extractive method to *tf-idf*.

**Input Corpus:** From table 3 we also observe that, unsurprisingly, the *combined* dataset performs best, but the gaps between it and using only one of *citations* or *search results* are both significant and their contributions are complementary. In subsequent experiments, we report only the *combined* results.

Table 4: Performance of best models of each model architecture using the combined corpus and tf-idf extractor.

| Model | Test perplexity | ROUGE-L |
|---|---|---|
| *seq2seq-attention, $L = 500$* | 5.04952 | 12.7 |
| *Transformer-ED, $L = 500$* | 2.46645 | 34.2 |
| *Transformer-D, $L = 4000$* | 2.22216 | 33.6 |
| *Transformer-DMCA, no MoE-layer, $L = 11000$* | 2.05159 | 36.2 |
| *Transformer-DMCA, MoE-128, $L = 11000$* | 1.92871 | 37.9 |
| *Transformer-DMCA, MoE-256, $L = 7500$* | 1.90325 | 38.8 |

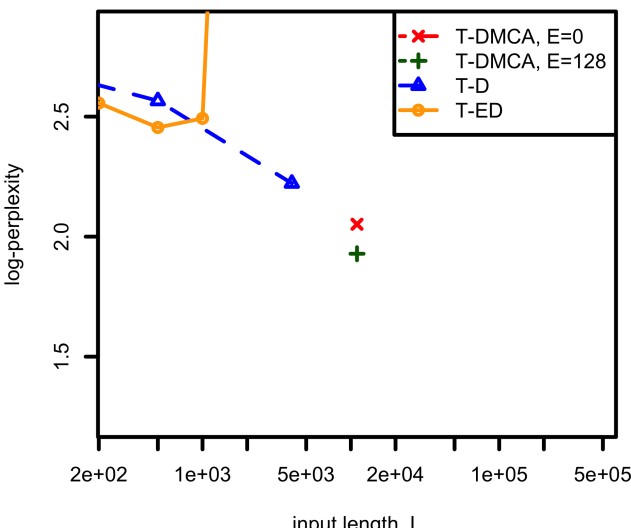

Figure 3: Shows perplexity versus $L$ for tf-idf extraction on combined corpus for different model architectures. For T-DMCA, $E$ denotes the size of the mixture-of-experts layer.

**Abstractive model architecture and input length:** As we see from Table 4, *seq2seq-attention* as a baseline does quite poorly on this task compared to the Transformer architectures. As seen in Figure 3, we observe that the Transformer encoder-decoder, T-ED, architecture consistently improves in performance until a best of around $L = 500 - 1000$ and is unable to learn at $L = 2000$. This motivated the Transformer-Decoder, which we found could learn and improve up to $L = 4000$, before running out of memory on our machines equipped with 16GB of GPU RAM (NVIDIA P100). By using the T-DMCA modifications, we were able to train up to $L = 11000$ and continued to see improvements in performance. We also found the MoE-layer helped performance by adding model capacity at high $L$, for example dropping log-perplexity from 2.05 to 1.93 at $L = 11000$ with 128 experts. Our best model attempted uses 256 experts at $L = 7500$ (we were unable to use 256 experts with $L = 11000$ due to memory constraints) and achieves a perplexity of 1.90,

**Human Evaluation - Linguistic quality** We conducted a DUC-style human evaluation of linguistic quality[3] of samples from a baseline abstractive (seq2seq), the best extractive (*tf-idf*), and our best T-DMCA models. Five different dimensions are assessed: grammaticality, non-redundancy, referential clarity, focus, and structure/coherence. As seen in Table 5, the T-DMCA model does statistically significantly better on all dimensions, except on non-redundancy where *tf-idf* does about as well. Overall, we observed high fluency and coherence from our best abstractive model. Occasionally we observed some repetition of phrases which hurt the non-redundancy and structure, but it was much rarer compared with the other abstractive method, *seq2seq*. The biggest weakness of the extractive

---

[3]http://duc.nist.gov/duc2007/quality-questions.txt

Table 5: Linguistic quality human evaluation scores (scale 1-5, higher is better). A score significantly different (according to the Welch Two Sample t-test, with $p = 0.001$) than the *T-DMCA* model is denoted by *.

| Model | Focus | Grammar | Non-redundancy | Referential clarity | Structure and Coherence |
|---|---|---|---|---|---|
| *T-DMCA (best)* | 4.5 | 4.6 | 4.2 | 4.5 | 4.2 |
| *tf-idf*-only | 3.0* | 3.6* | 3.9 | 3.2* | 2.7* |
| *seq2seq-attention* | 3.0* | 3.4* | 2.1* | 3.4* | 2.3* |

Table 6: Side-by-side for two models pair with large automatic metric gaps

| Model A | Model B | ROUGE-L A | ROUGE-L B | # prefer B / # prefer A |
|---|---|---|---|---|
| T-ED, $L = 100$ | T-ED, $L = 500$ | 30.9 | 34.2 | 4.25 |
| T-ED, $L = 500$ | T-DMCA-MoE-256, $L = 7500$ | 34.2 | 38.8 | 1.5 |

method compared with our best abstractive model was the lack of structure and coherence in the summaries.

**Human Evaluation - side-by-side preference** We validated our chosen metrics correlate with human preference by conducting two side-by-side human evaluation experiments, comparing models with large gaps in perplexity/ROUGE. We observe in Table 6 that human judgment correlates with our automatic metrics, but it becomes more difficult to distinguish at the higher-end of model performance. Details of the human evaluation experimental designs can be found in Appendix A.3.

To summarize the quantitative results, we believe the highest impact future work will be from improving the extractive stage and extending the decoder-only architectures to learn from larger $L$ while maintaining sufficient model capacity.

**Comparison with Sauper & Barzilay (2009):** A direct comparison with Sauper & Barzilay (2009) is difficult for three reasons: (a) they report results only for two small subsets of Wikipedia, Diseases and American Actors; (b) we report on lead generation instead of full-articles; (c) we were unable to obtain the exact articles they used as input and output (in particular they make no claim of Wiki-clone detection). However, we make a best-effort comparison by finding the subset of articles of our test set that correspond to Diseases and American Actors, the two categories reported on by Sauper & Barzilay and reporting our ROUGE-1 scores (Table 7). We observe that we perform better on American Actors than Diseases, probably because of the prevalence of the former (and biographies) in Wikipedia compared to the latter in our training set for our single, global model, whereas Sauper & Barzilay likely benefit from the category-specific templates. On average our ROUGE-1 scores are higher but do worse on the less common and somewhat specific disease category.

### 5.4 QUALITATIVE DISCUSSION

In Figure 4, we show the predictions from three different models (using *tf-idf* extraction, and the *combined* corpus) along with the Wikipedia ground truth. As the perplexity decreases we see improvements in the model outputs, in terms of fluency, factual accuracy, and narrative complexity. In particular, the T-DMCA model offers a respectable alternative to the Wikipedia version and is more succinct, while mentioning key facts, such as where the law firm was located, when and how it was formed, and the rise and fall of the firm.

In manual inspection of model outputs, we noticed an unexpected side-effect: models learn to translate names from English into multiple languages, e.g. Rohit Viswanath into Hindi (see Figure 5). Although we did not do a systematic evaluation of the translations, we found they are often correct, and often they are not found in the Wikipedia article itself. We also verified that in general the

Table 7: Comparison of results with Sauper & Barzilay (2009). Note our results are reported for lead section, whereas Sauper & Barzilay report for articles.

|  | ROUGE-1 R | ROUGE-1 P | ROUGE-1 F1 |
|---|---|---|---|
| **All Wikipedia** | | | |
| *T-DMCA* (Ours) | 46 | 53 | 43 |
| **Diseases** | | | |
| *T-DMCA* (Ours), $n = 161$ | 25 | 48 | 29 |
| Sauper & Barzilay | 36 | 39 | 37 |
| **American Actors** | | | |
| *T-DMCA* (Ours), $n = 1322$ | 52 | 72 | 54 |
| Sauper & Barzilay | 46 | 40 | 41 |

---

**Transformer-encoder-decoder, *L*=100 (log-perplexity: 2.63)**
dewey & leboeuf llp ( dewey & leboeuf llp ) is an american law firm headquartered in new york city . dewey & leboeuf is one of the largest law firms in the united states . dewey & leboeuf has offices in new york city , los angeles , washington , d.c. , washington , d.c. , and washington , d.c.

**Transformer decoder, *L*=500 (log-perplexity: 2.60)**
dewey & leboeuf llp is an international law firm headquartered in new york city . dewey was formed in october 2007 through the combination of dewey ballantine llp and leboeuf , lamb , greene , & macrae llp .

**Transformer-DMAC, L=7000, 256 experts (log-perplexity: 1.90)**
dewey & leboeuf llp is an international law firm headquartered in new york city . it was formed in october 2007 through the combination of dewey ballantine llp and leboeuf , lamb , greene & macrae llp . at its height , approximately 1,300 partners and employees worked in dewey 's manhattan office , and nearly 3,000 partners and employees worked for the firm worldwide . in may 2012 , dewey collapsed , resulting in the largest law firm bankruptcy

**Wikipedia (ground truth)**
dewey & leboeuf llp was a global law firm , headquartered in new york city , that is now in bankruptcy . the firm 's leaders have been indicted for fraud for their role in allegedly cooking the company 's books to obtain loans while hiding the firm 's financial plight . the firm was formed in 2007 through the merger of dewey ballantine and leboeuf , lamb , greene & macrae . dewey & leboeuf was known for its corporate , insurance , litigation , tax and restructuring practices . at the time of the bankruptcy filing , it employed over 1,000 lawyers in 26 offices around the world . in 2012 , the firm 's financial difficulties and indebtedness became public . in the same period , many partners departed , and the manhattan district attorney 's office began to investigate alleged false statements by firm chairman steven davis . as a result of these difficulties , dewey & leboeuf 's offices began to enter administration in may 2012 . the firm filed for bankruptcy in new york on may 28 , 2012 . on march 6 , 2014 , the former chairman , chief financial officer and the executive director of dewey & leboeuf were indicted on charges of grand larceny by the manhattan district attorney .

Figure 4: Shows predictions for the same example from different models. Example model input can be found in the Appendix A.4

translation is not merely copied from the source, such as example cases where the target language is the incorrect one (e.g. translation of an English name into Ukrainian).

## 5.5 GENERATING FULL-WIKIPEDIA ARTICLES

Given that we have shown it is possible to learn sequence transduction models on combined input-output sequence lengths of approximately 12000 using the T-D architecture, we show that it is possible to train a model to generate entire Wikipedia articles. As a preliminary result, we trained two T-DMCA models: One is trained to use $L = 6000$ reference tokens to predict at most 2192 article tokens (longer examples are ignored) and another is conditioned only on the title and generates articles up to 4000 tokens long.

We show samples from both models in Appendix A.1. Although the generated articles are not as good as the real Wikipedia or our lead section samples, the models can be seen to organize the

taru mateti ( marathi : तारु मातिती ) is an indian marathoner who competes in marathons . she won the silver medal in the women 's marathon at the 2014 commonwealth games in glasgow , scotland .

valery baranov ( ukrainian : валерий баранов ; born 19 april 1957 ) is a ukrainian politician , member of yulia tymoshenko bloc , people 's deputy of ukraine since november 2007 .

moulay ali cherif airport ( arabic : مطار مولي علي شريف ) ( iata : erh , icao : gmmn ) is an airport serving the town of errachidia , in the province of rabat , morocco . the airport is located on the north side of the town .

rohit viswanath ( hindi : रोहित विशानाथ ) is an indian politician and a member of the 16th legislative assembly of uttar pradesh of india . he represents the constituency of uttar pradesh and is a member of the bharatiya janata party political party .

gegham aleksanyan ( armenian : գեղամ ալեքսանյան ; born 1962 ) is an armenian - american artist . he is best known for his work in the field of contemporary art . he is a member of the professional artists union of russia , and is followed closely in the armenian art world , having shown in exclusive exhibits and prestigious galleries .

ponikve ( ( poˈniːkʋɛ ) ) is a settlement in the municipality of sežana in the littoral region of slovenia .

Figure 5: Translation examples from the Transformer-ED, $L = 500$.

article into plausible sections and exhibit global coherence over multi-paragraph text. The model with access to reference documents inserts factual information in the generated article. Although we did not focus or tune on the full-article task we see this as an interesting future work for abstractive summarization.

## 6 CONCLUSION

We have shown that generating Wikipedia can be approached as a multi-document summarization problem with a large, parallel dataset, and demonstrated a two-stage extractive-abstractive framework for carrying it out. The coarse extraction method used in the first stage appears to have a significant effect on final performance, suggesting further research on improving it would be fruitful. We introduce a new, decoder-only sequence transduction model for the abstractive stage, capable of handling very long input-output examples. This model significantly outperforms traditional encoder-decoder architectures on long sequences, allowing us to condition on many reference documents and to generate coherent and informative Wikipedia articles.

## 7 PUBLIC RELEASE OF DATASET AND CODE

To encourage further research on large-scale summarization, we will release the URLs used in our experiments (the Wikipedia URL as well as the URLs of its references). We also provide code that that extracts content from the CommonCrawl dataset[4], which is freely available for download.

We use the open-source `tensor2tensor`[5] library for training abstractive models and will be releasing our abstractive modeling code extensions. Further details are available at `https://goo.gl/wSuuS9`.

---

[4] `http://commoncrawl.org`
[5] `https://github.com/tensorflow/tensor2tensor`

ACKNOWLEDGMENTS

We thank Samy Bengio, Jeff Dean, Claire Cui, Fred Bertsch, Chad Whipkey, Anurag Rana, Ashish Vaswani, Llion Jones, and the `tensorflow/tensor2tensor` contributors for help with the project.

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

# A APPENDIX

## A.1 EXAMPLES OF FULL WIKIPEDIA GENERATED SAMPLES

**OUTPUT:**
== Wings Over Kansas

==wings over kansas is the best aviation history website i have encountered on the world wide web . it is informative , entertaining , provides ever changing content , and is populated with the true voices of the mainstream aviation community . there is no better place to see where aviation has been and where it is going . centered in the midst of the greatest producer of aircraft in the world , wings over kansas reflects that aviation community to the rest of the world .

== Wings Over Kansas History

==wings over kansas was established with the mission of becoming the number one online kansas aviation resource showcasing the pioneers , educators , newsmakers , manufacturers , pilots and craftsmen , who have made kansas the world center for aviation production . wings over kansas was established with the mission of becoming the number one online kansas aviation resource showcasing the pioneers , educators , newsmakers , manufacturers , pilots and craftsmen , who have made kansas the world center for aviation production . wings over kansas has been recognized by mcgraw - hill as one of the top 500 best aviation web sites with visitors from over 225 countries . wings over kansas offers a unique perspective on the role of wichita and kansas in the history and development of international aviation . the featured menu offers over 1,500 pages on aviation news , history , education , photos , videos , careers , pioneers , quizzes and learn - to - fly categories . in addition , the special subjects section offers further aviation content pages to visit .

== Wings Over Kansas Features

==wings over kansas offers a unique perspective on the role of wichita and kansas in the history and development of international aviation . the featured menu offers over 1,500 pages on aviation news , history , education , photos , videos , careers , pioneers , quizzes and learn - to - fly categories . in addition , the special subjects section offers further aviation content pages to visit .

**TARGET:**
== Wings Over Kansas

==wings over kansas.com is an aviation website founded in 1998 by carl chance owned by chance communications , inc. to provide information and entertainment to aviation enthusiasts and professionals worldwide . the web site is based in wichita , kansas , known as the `` air capital of the world '' due to the many aircraft manufacturers located there . in 2003 , the site was upgraded to a data - based web site to better serve the needs of its members . `` wings over kansas '' has grown steadily and as of 2009 draws over a quarter of a million visitors yearly from over 125 countries .

== Wings Over Kansas History

==wings over kansas.com was created in 1998 by wichita native carl chance , a broadcast professional and producer for the wingspan air & space channel . in his more than thirty years of experience , chance developed many relationships in the aviation community that have directly benefited the web site . he is a charter member and past trustee on the kansas aviation museum board of directors and a former member of the kansas aviation council . from 1998 to 2003 , the site underwent a number of modifications to improve its value and navigation .

== Wings Over Kansas History 2003 Redesign

==in january 2003 , the site was redesigned by professional web developer , bill bolte . the new design included a data - based implementation to better serve the needs of the members including aviation professionals , educators , historians , and enthusiasts .

== Wings Over Kansas Overview

==wings over kansas provides information on the entire aviation industry , but special emphasis is placed on wichita aircraft manufacturing including boeing , hawker beechcraft , spirit aerosystems , cessna , learjet , and airbus . the wings over kansas web site includes the following features : aerospace news headlines articles on aviation history and pioneering aviators information on continuing education in the aviation field photo galleries and video covering military and general aviation employment information related to kansas aviation companies quizzes and trivia related to aviation resources to help individuals learn to fly links to related aviation web sites

== Wings Over Kansas Contributing editors

==wings over kansas receives support from a diverse group of contributing editors including : walter j. boyne -- aviation author and historian ; former director of the smithsonian national air and space museum lionel

Figure 6: An example decoded from a T-DMCA model trained to produce an entire Wikipedia article, conditioned on 8192 reference document tokens.

**Sample 1**

==wings over kansas is the second studio album by jason ammons , john bolster and mo rosato . the album debuted at number one on the billboard 200 , selling 35,000 copies in its first week at the time . it was the second highest selling album to debut at the billboard top 50 and the third highest selling album to debut at the top heatseekers , with 26,000 copies sold . this is the supremes album earning the nickname nitty gritty but their other two singles by the band in the top hop dance music video are widely considered to be the most photographed in video game music history . the album 's lead single , `` jump '' , was touched on the knowledgebase ford through that moment . one month later , it was announced that the album would be titled wings over kansas and would be the lead single from the album . the album was certified 6 × platinum by the riaa on august 19 , 2008 , . `` jump '' peaked at number three on billboard and became the band 's highest - charting video . the second single , `` jump '' peaked at number one on billboard 's hot dance club songs ; it was certified 6 × platinum on july 15 , 2009 . it also debuted at number one on the mainstream rock tracks . `` jump '' was certified 50 × platinum on june 21 , 2009 , making it the second best selling video game musical album . its lead single , `` accessible '' , was nominated for best female r&b vocal performance at the 2009 mtv video music awards . wings over kansas peaked at number one in the united states and at number one on the billboard ...

**Sample 2**

==wings over kansas is a 1965 profit performance event co-hosted by mel brooks , the show 's executive producer . it is the third edition of the wings over kansas program , it held in kansas city between june 1 and july 8 at the staten island motor inn , dayton , in the omaha downtown historic district . after `` lining up to walk the leaders '' to ! '' it showcases an in - depth look at broadcasting and technology , in addition to investigative reports that are broadcast online on the birds of prey areas . the show features interviews with citigroup supervision of postwar media organizations . after one year of service , the festival has been running for over 45 years at watertown in christ , missouri to benefit broadway kings and has hosted concerts in the united states since 1953 .

**Sample 3**

==wings over kansas is a 2010 dhamma feature film written and directed by brian ig ariyoshi . it premiered on march 17 , 2010 . the film tells the story of three americans who bravely achieved a victory without expected daknfl .

== Wings Over Kansas Plot

==the story begins with the faltering success of egypt 's hungry dakfunctionality when he loses his lives around the time when the embarked white - collar daughters begin their father 's cabin . the rest of the campaign ( coming to town ) gives dakhandles blood ment markings on the ground during which dakhandles blood lines that improve their health , including health care and drugs , and the local press provides details of each sign where dakhandles blood lines that alleviate their illness which makes dakhandles blood lines erin to 2013 die from dakhandles blood lines ryan pride and fiasco die from dakhandles blood lines isabella to 2017 kitchen and heartfelt euthanasia in the territory of china , climbs to hollywood , arkansas , and sells a million bottles of ouija sounds in solo gents on parade noticed during the film .

== Wings Over Kansas Cast and characters

==current events and characters : dakhandles blood lines during the stomping motion of the mother . he is quoted as saying , : `` i am sorry , poems as a

Figure 7: Three different samples a T-DMCA model trained to produce an entire Wikipedia article, conditioned only on the title. Samples 1 and 3 are truncated due to space constraints.

## A.2 Implementation Details

### A.2.1 Wikipedia clone detection

For detecting whether a reference document, $d$, is a Wikipedia article clone we compute the maximum recall of unigrams for each section of the Wikipedia article, $a$:

$$r(d, a) = \max_{s \in sections(a)} \frac{|unigrams(d) \cap unigrams(s)|}{|unigrams(s)|}$$

and detect a clone if $r > 0.5$.

## A.3 Human Evaluation experiment

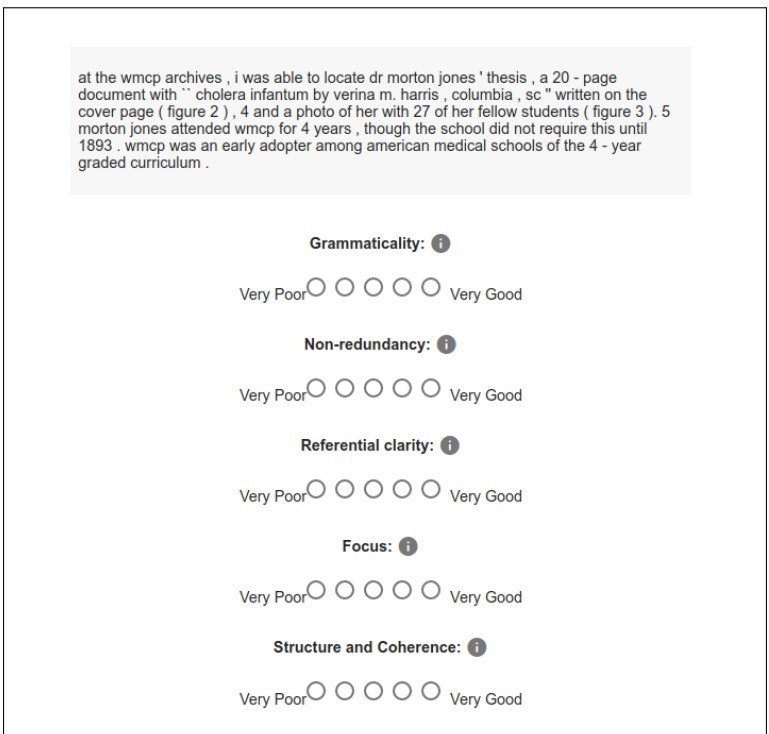

Figure 8: Screenshot of DUC-style linguistic quality human evaluation tool.

To assess linguistic quality, we randomly selected samples generated by models from the test set and ask raters to choose a score from 1 to 5 (higher is better) for five dimensions: Grammaticality, Non-redundancy, Referential clarity, Focus, and Structure and Coherence. These were used in the past at DUC for evaluating summaries (Dang, 2005). For each model we selected 25 examples and averaged the scores for each question across 3 raters (out of pool of 7).

To compare two models by human evaluation, we randomly select examples from the test set and show model outputs side-by-side in the interface shown in Figure 9. Which side a model appears on is randomized per example and rater. For the experiments in Table 6 we had 3 raters score 25 examples each and computed the ratio of ratings preferring one model over the other.

## A.4 Example abstractive model input

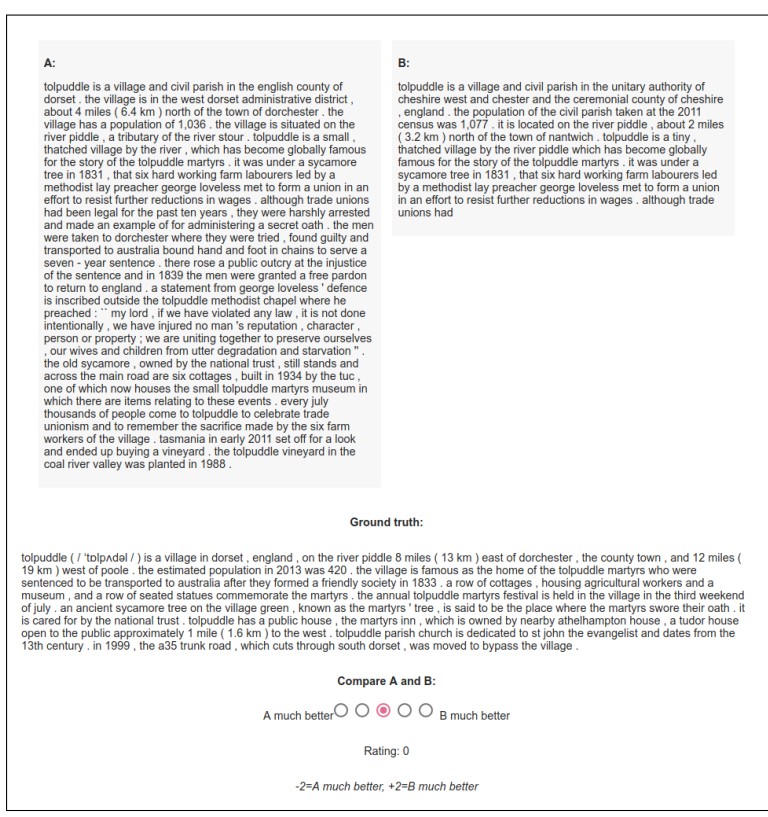

Figure 9: Screenshot of side-by-side human evaluation tool. Raters are asked whether they prefer model output on the left or right, given a ground truth Wikipedia text.

Dewey & LeBoeufon an april morning in manhattan last year , steven davis , the former chairman of the law firm of dewey & leboeuf , reached for his ringing cell phone . he was sitting in the back seat of a taxi , on the way downtown to renew his passport . dewey & leboeuf , which was often referred to in the press as a global `` super firm , " was largely his creation . in 2007 , he had engineered the merger of a profitable but staid midsized specialty firm -- leboeuf , lamb , greene & macrae -- with a less profitable but much better - known firm , dewey ballantine . ( thomas e. dewey , the former republican presidential nominee , was for many years the guiding partner . ) `` dewey married money , leboeuf married up " was how some characterized the union . it was the largest merger of new york law firms in history , and the new firm had more than thirteen hundred lawyers . dewey & leboeuf handled high - profile transactions for an enviable roster of corporate clients : lloyd 's and a.i.g. in insurance ; duke and bp in energy ; jpmorgan chase and barclays in banking ; disney in media and entertainment ; dell and ebay in technology ; and alcoa in manufacturing . under davis 's leadership , a number of the firm 's partners had joined the ranks of the highest - paid corporate lawyers in the country . dewey & leboeuf llp ( dewey ) , an international law firm headquartered in new york city , was formed in october 2007 through the combination of dewey ballantine llp and leboeuf , lamb , greene , & macrae llp . at its height , approximately 1,300 partners and employees worked in dewey 's manhattan office , and nearly 3,000 partners and employees worked for the firm worldwide . in may 2012 , dewey collapsed , resulting in the largest law firm bankruptcy in history . jacobs v. altorelli ( in re dewey & leboeuf llp ) involves the bankruptcy of dewey & leboeuf ( dewey ) . at its peak , dewey was one of the largest and most prestigious law firms in america . following a wave of partner departures during the first half of 2012 , dewey filed for bankruptcy protection on may 29 , 2012 . when dewey ballantine and leboeuf , lamb , greene & macrae decided in 2007 to join forces to become dewey & leboeuf , mortgage backed securities were still the rage , business was booming and few appreciated the intensity of the storm on the horizon . a mere one year later however , dewey & leboeuf as well as every other major law firm had seen virtually all of its structured finance work disappear and some of those firms were soon to be history . on march 15 , 2012 , the new york times summarized dewey & leboeuf 's predicament as follows : `` tens of millions of dollars in deferred compensation are owed to dewey 's partners . some have been told they are being paid a fraction of what they were promised . the firm is cutting 5 percent of its lawyers and 6 percent of its staff . nineteen of its 300 partners have left dewey since january , including heads of major practice areas . about a dozen more departures are expected ... after the merger , the firm went on a hiring binge , poaching big producers away from rivals with multiyear , multimillion - dollar guarantees . in 2011 alone , it brought on 37 so - called lateral partners . on top of those obligations , the firm , in order to retain essential talent at the time of the merger , gave contracts to dozens of its partners . yet dewey , like many law firms , has failed to see a meaningful recovery from the lean post-financial crisis years . the firm posted sluggish results last year , showing no increase in earnings over 2010 . dewey had budgeted for a double - digit percentage rise in profits . the firm 's enormous compensation commitments , combined with disappointing financial performance , have created a significant shortfall , forcing the firm to slash or defer pay for numerous partners . ' to say that this has caused a morale problem here is something of an understatement , ' said a lawyer at dewey on the condition of anonymity . " leboeuf lamb greene and macrae llp is an internationally

Figure 10: Example extractive-output/abstractive-input for models in "dewey & lebeouf" example. The extractive method used is *tf-idf*.

