# OpenReview forum: "Generating Wikipedia by Summarizing Long Sequences"
_ICLR.cc/2018/Conference — Accept (Poster)_

### Official Review · AnonReviewer3 · 2017-11-24
**Interesting, but evaluation could have been stronger**

**Rating:** 7
**Confidence:** 5

**Review:**

This paper considers the task of generating Wikipedia articles as a combination of extractive and abstractive multi-document summarization task where input is the content of reference articles listed in a Wikipedia page along with the content collected from Web search and output is the generated content for a target Wikipedia page. The authors at first reduce the input size by using various extractive strategies and then use the selected content as input to the abstractive stage where they leverage the Transformer architecture with interesting modifications like dropping the encoder and proposing alternate self-attention mechanisms like local and memory compressed attention.

In general, the paper is well-written and the main ideas are clear. However, my main concern is the evaluation. It would have been nice to see how the proposed methods perform with respect to the existing neural abstractive summarization approaches. Although authors argue in Section 2.1 that existing neural approaches are applied to other kinds of datasets where the input/output size ratios are smaller,  experiments could have been performed to show their impact. Furthermore, I really expected to see a comparison with Sauper & Barzilay (2009)'s non-neural extractive approach of Wikipedia article generation, which could certainly strengthen the technical merit of the paper.

More importantly, it was not clear from the paper if there was a constraint on the output length when each model generated the Wikipedia content. For example, Figure 5-7 show variable sizes of the generated outputs. With a fixed reference/target Wikipedia article, if different models generate variable sizes of output, ROUGE evaluation could easily pose a bias on a longer output as it essentially counts overlaps between the system output and the reference.

It would have been nice to know if the proposed attention mechanisms account for significantly better results than the T-ED and T-D architectures. Did you run any statistical significance test on the evaluation results?

Authors claim that the proposed model can generate "fluent, coherent" output, however, no evaluation has been conducted to justify this claim. The human evaluation only compares two alternative models for preference, which is not enough to support this claim. I would suggest to carry out a DUC-style user evaluation (http://www-nlpir.nist.gov/projects/duc/duc2007/quality-questions.txt) methodology to really show that the proposed method works well for abstractive summarization.

Does Figure 8 show an example input after the extractive stage or before? Please clarify.

---------------
I have updated my scores as authors clarified most of my concerns.

---

> ### Author Response · Authors · 2017-12-19
> **We agree evaluation needed improvement and believe we have addressed your feedback**
>
> “It would have been nice to see how the proposed methods perform with respect to the existing neural abstractive summarization approaches. Although authors argue in Section 2.1 that existing neural approaches are applied to other kinds of datasets where the input/output size ratios are smaller,  experiments could have been performed to show their impact.”
>
> - In addition to our proposed neural architectures, we compare to very strong existing baselines, seq2seq with attention, which gets state-of-the-art on the Gigaword summarization task as well as the Transformer encoder-decoder, which gets state-of-the-art on translation, a related task from which most summarization techniques arise. We show our models significantly outperform those competitive abstractive methods.
>
>
> “Furthermore, I really expected to see a comparison with Sauper & Barzilay (2009)'s non-neural extractive approach of Wikipedia article generation, which could certainly strengthen the technical merit of the paper.”
>
> - This is a fair point and we added a section comparing with Sauper & Barzilay in Experiments.
>
>
> “More importantly, it was not clear from the paper if there was a constraint on the output length when each model generated the Wikipedia content. For example, Figure 5-7 show variable sizes of the generated outputs. With a fixed reference/target Wikipedia article, if different models generate variable sizes of output, ROUGE evaluation could easily pose a bias on a longer output as it essentially counts overlaps between the system output and the reference.”
>
> - The models are not constrained to output a certain length. Instead we generate until an end-of-sequence token is encountered. There is a length-penalty, alpha, that we tune based on performance of the validation set. In our case, the ROUGE F1 evaluation is fair because it is the harmonic mean of ROUGE-Recall (favors long summaries) and ROUGE-Precision (favors short summaries). As a result, longer output is penalized if it is not useful and related to the target. We tried to clarify this in the Experiments section.
>
>
> “It would have been nice to know if the proposed attention mechanisms account for significantly better results than the T-ED and T-D architectures.”
>
> - We don’t claim for this task that T-D does better than T-ED for short input lengths. However, we hope Figure 3 makes it clear that the T-ED architecture begins to fail and is no longer competitive for longer inputs. In particular, our architecture improvements allow us to consider much larger input lengths, which results in significantly higher ROUGE and and human evaluation scores.
>
>
> “Authors claim that the proposed model can generate "fluent, coherent" output, however, no evaluation has been conducted to justify this claim. The human evaluation only compares two alternative models for preference, which is not enough to support this claim. I would suggest to carry out a DUC-style user evaluation (http://www-nlpir.nist.gov/projects/duc/duc2007/quality-questions.txt) methodology to really show that the proposed method works well for abstractive summarization.”
>
> - This is a good point and we followed your suggestion and added a DUC-style human evaluation of linguistic quality. We hope we make it clear that the best abstractive model proposed is significanlty much more fluent/coherent than the best extractive method we tried and another baseline abstractive method (seq2seq). The quality scores are also high in the absolute sense.
>
>
> “Does Figure 8 show an example input after the extractive stage or before? Please clarify.”
>
> - We clarified in the paper (now Figure 10) that it is the output of the extractive stage, before the abstractive stage.

---

### Official Review · AnonReviewer1 · 2017-11-27
**Impressive application, but hard to judge technical contribution**

**Rating:** 8
**Confidence:** 3

**Review:**

This paper proposes an approach to generating the first section of Wikipedia articles (and potentially entire articles).
First relavant paragraphs are extracted from reference documents and documents retrieved through search engine queries through a TD-IDF-based ranking. Then abstractive summarization is performed using a modification of Transformer networks (Vasvani et al 2017). A mixture of experts layer further improves performance.
The proposed transformer decoder defines a distribution over both the input and output sequences using the same self-attention-based network. On its own this modification improves perplexity (on longer sequences) but not the Rouge score; however the architecture enables memory-compressed attention which is more scalable to long input sequences. It is claimed that the transformer decoder makes optimization easier but no complete explanation or justification of this is given. Computing self-attention and softmaxes over entire input sequences will significantly increase the computational cost of training.

In the task setup the information retrieval-based extractive stage is crucial to performance, but this contribution might be less important to the ICLR community. It willl also be hard to reproduce without significant computational resources, even if the URLs of the dataset are made available. The training data is significantly larger than the CNN/DailyMail single-document summarization dataset.

The paper presents strong quantitative results and qualitative examples. Unfortunately it is hard to judge the effectiveness of the abstractive model due to the scale of the experiments, especially with regards to the quality of the generated output in comparison to the output of the extractive stage.
In some of the examples the system output seems to be significantly shorter than the reference, so it would be helpful to quantify this, as well how much the quality degrades when the model is forced to generate outputs of a given minimum length. While the proposed approach is more scalable, it is hard to judge the extend of this.

So while the performance of the overall system is impressive, it is hard to judge the significance of the technical contribution made by the paper.

---
The additional experiments and clarifications in the updated version give substantially more evidence in support of the claims made by the paper, and I would like to see the paper accepted.

---

> ### Author Response · Authors · 2017-12-19
> **Hopefully the technical contribution is a lot clearer after significantly augmented evaluation**
>
> Thank you for the detailed review with actionable feedback. We found common feedback from the three reviewers to augment the evaluation section of the paper and believe we have significantly improved it. In particular, please see responses below in-line as well as rebuttals for other reviews and the summary of changes above.
>
>
> “In the task setup the information retrieval-based extractive stage is crucial to performance, but this contribution might be less important to the ICLR community.”
>
> - We added additional analysis to demonstrate that the extractive stage while important, is far from sufficient to produce good wikipedia articles. We show that ROUGE and human evaluation are greatly improved by the abstractive stage.
>
>
> “It willl also be hard to reproduce without significant computational resources, even if the URLs of the dataset are made available.”
>
> - We will be providing a script for generating the dataset from the CommonCrawl dataset (which is freely available for download). It will run locally instead of downloading over the Internet, and so will be relatively much faster.
>
>
> “Unfortunately it is hard to judge the effectiveness of the abstractive model due to the scale of the experiments, especially with regards to the quality of the generated output in comparison to the output of the extractive stage.”
>
> - We added analysis of the incremental performance of the abstractive model over the extractive output in terms of ROUGE and human evaluation (DUC-style linguistic quality evaluation). We believe we make a strong case that the abstractive model is doing something highly non-trivial and significant.
>
>
> “In some of the examples the system output seems to be significantly shorter than the reference, so it would be helpful to quantify this, as well how much the quality degrades when the model is forced to generate outputs of a given minimum length.”
> - We clarified in the paper that the models are not constrained to output a certain length. Instead we generate until an end-of-sequence token is encountered. There is a length-penalty, alpha, that we tune based on performance of the validation set.
>
>
> “So while the performance of the overall system is impressive, it is hard to judge the significance of the technical contribution made by the paper.”
> - In addition to the proposed task/dataset, we believe the technical significance is demonstrating how very long text-to-text sequence transduction tasks can be done. Previous related work in translation or summarization focused on much shorter sequences. We had to introduce new model architectures to solve this new problem and believe it would be of great interest to the ICLR community. We hope our added evaluations make this claim more convincing.

---

> > ### Comment · AnonReviewer1 · 2018-01-01
> > **Thanks for your detailed response**
> >
> > Thanks for your detailed response and follow-up work!

---

### Official Review · AnonReviewer2 · 2017-11-28
**Interesting task. Would appreciate more analysis.**

**Rating:** 7
**Confidence:** 4

**Review:**

The main significance of this paper is to propose the task of generating the lead section of Wikipedia articles by viewing it as a multi-document summarization problem. Linked articles as well as the results of an external web search query are used as input documents, from which the Wikipedia lead section must be generated. Further preprocessing of the input articles is required, using simple heuristics to extract the most relevant sections to feed to a neural abstractive summarizer. A number of variants of attention mechanisms are compared, including the transofer-decoder, and a variant with memory-compressed attention in order to handle longer sequences. The outputs are evaluated by ROUGE-L and test perplexity. There is also a A-B testing setup by human evaluators to show that ROUGE-L rankings correspond to human preferences of systems, at least for large ROUGE differences.

This paper is quite original and clearly written. The main strength is in the task setup with the dataset and the proposed input sources for generating Wikipedia articles. The main weakness is that I would have liked to see more analysis and comparisons in the evaluation.

Evaluation:
Currently, only neural abstractive methods are compared. I would have liked to see the ROUGE performance of some current unsupervised multi-document extractive summarization methods, as well as some simple multi-document selection algorithms such as SumBasic. Do redundancy cues which work for multi-document news summarization still work for this task?

Extractiveness analysis:
I would also have liked to see more analysis of how extractive the Wikipedia articles actually are, as well as how extractive the system outputs are. Does higher extractiveness correspond to higher or lower system ROUGE scores? This would help us understand the difficulty of the problem, and how much abstractive methods could be expected to help.

A further analysis which would be nice to do (though I have less clear ideas how to do it), would be to have some way to figure out which article types or which section types are amenable to this setup, and which are not.

I have some concern that extraction could do very well if you happen to find a related article in another website which contains encyclopedia-like or definition-like entries (e.g., Baidu, Wiktionary) which is not caught by clone detection. In this case, the problem could become less interesting, as no real analysis is required to do well here.

Overall, I quite like this line of work, but I think the paper would be a lot stronger and more convincing with some additional work.

----
After reading the authors' response and the updated submission, I am satisfied that my concerns above have been adequately addressed in the new version of the paper. This is a very nice contribution.

---

> ### Author Response · Authors · 2017-12-19
> **We significantly augmented the evaluation section and tried addressing all your feedback**
>
> Thank you for the detailed review with actionable feedback. We found common feedback from the three reviewers to augment the evaluation section of the paper and believe we have significantly improved it. In particular, please see responses below in-line where we address all of your feedback.
>
> “This paper is quite original and clearly written. The main strength is in the task setup with the dataset and the proposed input sources for generating Wikipedia articles. “
>
> - In addition to the task setup, we believe we’ve demonstrated how to do very long (much longer than previously attempted) text-to-text sequence transduction and introduced a new model architecture to do it. We believe this is of great interest to the ICLR community.
>
> “Currently, only neural abstractive methods are compared. I would have liked to see the ROUGE performance of some current unsupervised multi-document extractive summarization methods, as well as some simple multi-document selection algorithms such as SumBasic. Do redundancy cues which work for multi-document news summarization still work for this task?”
>
> - We implemented SumBasic and TextRank (along with tf-idf) to evaluate extractive methods on their own and evaluated them on this task. I believe we show convincingly in the results (e.g. extractive bar-plot) that the abstractive stage indeed adds a lot to the extractive output in terms of ROUGE and human evaluation of linguistic quality and that redundancy cues are not enough.
>
> “I would also have liked to see more analysis of how extractive the Wikipedia articles actually are, as well as how extractive the system outputs are. Does higher extractiveness correspond to higher or lower system ROUGE scores? This would help us understand the difficulty of the problem, and how much abstractive methods could be expected to help.”
>
> - In Section 2.1 we computed the proportion of unigrams/words in the output co-occurring in the input for our task and for the Gigaword and CNN/DailyMail datasets and showed that by this measure WikiSum is much less extractive. In particular, the presence of wiki-clones in the input would give a score of 100%, whereas we report 59.2%.
>
> “A further analysis which would be nice to do (though I have less clear ideas how to do it), would be to have some way to figure out which article types or which section types are amenable to this setup, and which are not.”
>
> - We added a comparison in the paper with Sauper & Barzilay on two Wiki categories. It turns out we do worse on Diseases compared to Actors. We think this is because we use a single model for all categories and the training data is heavily biased toward people.
>
> “I have some concern that extraction could do very well if you happen to find a related article in another website which contains encyclopedia-like or definition-like entries (e.g., Baidu, Wiktionary) which is not caught by clone detection. In this case, the problem could become less interesting, as no real analysis is required to do well here.”
> - We hope our added analysis in Section 2.1 mentioned above should address this concern.

---

### Author Response · Authors · 2017-12-19
**Summary of changes to revision 12/18/2017**

- We found common, actionable feedback from the three reviewers to augment the evaluation section of the paper and believe we have significantly improved it.
    - We added results from a DUC-style linguistic quality human evaluation, showing our model significantly outperforms pure extractive methods we tried as well as an abstractive baseline.
    - We quantified the performance of extractive-only methods on our proposed task, adding results of two more well-cited methods, SumBasic and TextRank (in addition to tf-idf). We show that the abstractive stage indeed adds significant lift to ROUGE and human evaluation performance.
    - We added a section with a comparison with Sauper and Barzilay.

- We quantify the extractiveness of the dataset in section 2.1. Although we mentioned we had a Wiki clone-detection algorithm, we didn’t quantify the results. In Table 1, we show that the proportion of unigrams/words in the output co-occurring in the input is much lower than in other summarization datasets at 59.2%.

- Some other clarifications made in paper
    - Modified caption for Figure 8 to make it clear it is the output of the extractive stage and what the abstractive model uses as input for article generation.
    - We note output length is not constrained in abstractive models. There’s a length penalty hyper-parameter, \alpha, that is tuned on the validation set. For ROUGE-L scores, we report the more appropriate F1 flavor, which is the harmonic
mean of ROUGE-Recall (favors long summaries) and ROUGE-Precision (favors short summaries).
elaborated on justification for T-D architecture for monolingual text-to-text problems

- On feasibility of reproducibility: We will be providing a script that generates the data locally from a local copy of the CommonCrawl dataset, which can be downloaded from http://commoncrawl.org/.  Because the script will run locally it will be significantly faster than downloading webpages from the Internet.

Overall we believe the paper is much stronger after the suggestions from the reviewers. Please re-consider scores after this revision. Thank you!

---

### Public Comment · ~Bella_Thomas1 · 2020-07-07
**Creating Wikipedia content, what to avoid**

In order to create an exceptional and impactful Wikipedia content it is important that you integrate effective strategies. Ones that are going to make a positive influence on the viewer and help you convey your messages across. You need to make sure to <a href="https://makeawikipage.org/">make a Wikipedia page</a>.  that is going to be well researched and thorough on the context of your content. This will make it easier for your viewers to connect with. Make sure you avoid biases and misinformation. Do you think there are any other factors that should be avoided to make a Wikipedia page?

---

### Decision · Program_Chairs · 2018-01-29
**ICLR 2018 Conference Acceptance Decision**

**Decision:**

Accept (Poster)

**Comment:**

This paper presents a new multi-document summarization task of trying to write a wikipedia article based on its sources. Reviewers found the paper and the task clear to understand and well-explained. The modeling aspects are clear as well, although lacking justification. Reviewers are split on the originality of the task, saying that it is certainly big, but wondering if that makes it difficult to compare with. The main split was the feeling that "the paper presents strong quantitative results and qualitative examples. " versus a frustration that the experimental results did not take into account extractive baselines or analysis. However the authors provide a significantly updated version of the work targeting many of these concerns, which does alleviate some of the main issues. For these reasons, despite one low review, my recommendation is that this work be accepted as a very interesting contribution.